# Plasma Cells as the Key Players of IVF Failure? Unlocking the Enigma of Infertility and In Vitro Fertilization Failure in the Light of Uterine Inflammation

**DOI:** 10.3390/ijms252313083

**Published:** 2024-12-05

**Authors:** Ewa Dwojak, Magdalena Mroczek, Grzegorz Dworacki, Paula Dobosz, Antonina Ślubowska, Maria Stępień, Martyna Borowczyk, Izabela Filipczyńska, Agata Tomaszewska, Rafał Ałtyn, Hanna Chowaniec

**Affiliations:** 1Department of Pathomorphology and Clinical Immunology, Poznan University of Medical Sciences, 60-355 Poznan, Poland; gdwrck@gmail.com (G.D.);; 2Department of Pathomorphology, Poznan University Hospital, 60-355 Poznan, Poland; 3Department of Neurology, University Hospital Basel, 4031 Basel, Switzerland; 4Department of Biostatistics and Research Methodology, Collegium Medicum, Faculty of Medicine, Cardinal Stefan Wyszynski University of Warsaw, 01-938 Warsaw, Poland; 5Université Paris-Saclay, UVSQ, INSERM END-ICAP, 78000 Versailles, France; 6Doctoral School, Medical University of Lublin, 20-059 Lublin, Poland; 7Department of Endocrinology, Metabolism and Internal Medicine, Poznan University of Medical Sciences, 60-355 Poznan, Poland; martyna.borowczyk@gmail.com; 8Department of Reproduction and Gynecological Endocrinology, Medical University of Bialystok, 15-276 Bialystok, Poland; 9The Center for Medical Genetics GENESIS, 60-406 Poznan, Poland; 10University Center for Cancer Diagnostics, Poznan University of Medical Sciences, 60-806 Poznan, Poland; 11University Center for Clinical Research Support, Poznan University of Medical Sciences, 60-354 Poznan, Poland; 12Information Technology Department, Poznan University of Medical Sciences, 60-806 Poznan, Poland

**Keywords:** endometrial inflammation, plasma cell, CD138, syndecan-1, MUM1/IRF-4, in vitro fertilisation, endometrium, chronic endometritis

## Abstract

There is an interplay between plasma cells, endometritis, and infertility, particularly in the context of in vitro fertilization (IVF) failure. This narrative literature review explains the pathophysiology of endometritis, detailing the involvement of various immune cells, cytokines, and chemokines in the regulation of inflammatory responses within the uterine endometrium. Here, we discuss the physiological role of plasma cells in immunity and their detection as markers of chronic endometritis, a disease associated with reproductive disorders. Our study also highlights the importance of CD138 immunohistochemical staining in the diagnosis of chronic endometritis, emphasizing the presence of plasma cells in endometrial tissue and its association with infertility and recurrent implantation failure. Of particular interest are the proposed diagnostic criteria for chronic endometritis based on the presence of plasma cells and studies that suggest a threshold for diagnosing this condition. We highlight the importance of examining the regenerative potential of endometrial stem cells in the treatment of infertility related to endometrial disorders.

## 1. Introduction

Recently, the role of the plasma cells in endometriosis and chronic endometritis has been emphasized, especially in the context of infertility and IVF failure. Experimental studies showed that, indeed, counting plasma cells in endometrial biopsies may be a new biomarker for endometriosis and miscarriage rates [1,2]. Given the association between miscarriages, endometriosis, and plasma cells, CD138 immunochemistry in endometrial tissue can be recommended for miscarriages and infertility [3]. In this narrative review, we analyzed the role of the plasma cells as a biomarker in endometriosis and its association with miscarriages and IVF failure.

Plasma cells are terminally differentiated white blood cells, B cells, that produce and secrete antibodies [4]. They play an important role in providing long-term immunity through the production of memory B cells, which can rapidly generate large quantities of specific antibodies upon re-exposure to the same antigen. Deviation in their differentiation may lead to a variety of disorders, such as Monoclonal Gammopathy of Unidentified Significance (MGUS), Smouldering Myeloma, Plasma Cell Myeloma (PCM), and over 100 immune deficiencies [5]. On the ultrastructural level, plasma cells are characterized by eccentric round nuclei (Nu), the cytoplasm is rich in rough endoplasmic reticulum (RER) arranged in parallel arrays, cytoplasmic fibrils, and numerous polymorphic mitochondria (Mi)—as depicted on Figure 1 below. They are also characterized by the inclusions, ranging from elliptical to round or needle-like crystalline structures, that primarily consist of accumulated immunoglobulins [6].

Physiologically, plasma cells are located in the bone marrow, spleen, and Mucosal Associated Lymphoid Tissue (MALT), mainly in the lamina propria of mucosal membranes in the gastrointestinal tract and respiratory tract [7]. An excellent marker for the evaluation of plasma cells is the expression of CD138 (Syndecan-1) [8]. Using the CD138 staining, it was proven that plasma cells are present in chronic endometritis and are hallmarks of the inflammation of the endometrium [3,9]. Plasma cells can be seen as a potential biomarker in endometritis, although there is no clear threshold defined at this moment [10]. The diagnosis of chronic endometritis is based basically on the hysterectomy with biopsy and the presence of plasma cells in the endometrium [2,10]. In addition, endometritis with the presence of plasma cells has been proven to occur in women diagnosed with infertility, recurrent implantation failure, and intracytoplasmic sperm injection failure [11,12]. The high rate of chronic endometritis was also observed in cases of retained pregnancy tissue after miscarriage [13]. The presence of plasma cells has also been observed in women diagnosed with endometrial polyps [14,15].

Another excellent marker for staining plasma cells is multiple myeloma 1/interferon regulatory factor 4 (MUM1/IRF-4). Apart from plasma cells, it is also expressed in melanocytes, activated B cells, and activated T cells [16]. Since MUM1/IRF-4 has a nuclear expression, simultaneously labeling MUM1/IRF4 and CD138 (which is a surface cell marker) improves the accuracy and efficiency of chronic endometritis diagnostics, but even CD138 alone tends to show nonspecific` staining on epithelial cells [17].

## 2. Results and Discussion

### 2.1. Assessing Uterine Endometrium with Transvaginal Ultrasonography

The endometrium, the inner lining of the uterus, undergoes cyclical changes throughout the menstrual cycle in response to fluctuations in hormone levels. These changes are crucial for preparing the uterus for potential pregnancy and, if pregnancy doesn’t occur, for the shedding of the endometrial lining during menstruation [18]. These cyclical changes in the endometrium are orchestrated by the complex interplay of estrogen, progesterone, and other hormones, regulated by feedback mechanisms involving the hypothalamus, pituitary gland, and ovaries. These changes ensure the cyclical renewal and preparation of the endometrium for potential pregnancy each month. High-resolution transvaginal sonography (TVS) makes it possible to monitor noninvasively the endometrium through TVS assessment of the pattern of the endometrium wherein the endometrial echogenicity is assessed [19]. Patterns of endometrium can be graded as Type A-hypoechoic/trilaminar (hypoechoic endometrium with prominent central and outer echogenic lines), Type B-isoechoic (same reflectivity of the endometrium as the surrounding myometrium and a poorly defined central echogenic line), and Type C-hyperechoic (increased reflectivity compared to myometrium and the central and the outer echogenic line not differentiated clearly [20]. It is well established that in the proliferative phase, the endometrium has a triple line/hypoechoic endometrium (Type A), and this appearance changes in the secretory phase, becoming hyperechoic (Type C) [19].

R. Narayan and R. K. Gosway proved in their work that transvaginal sonography can be successfully used to confirm the presence of endometritis [21]. The result obtained in this examination may constitute the basis for further tests aimed at confirming or denying the presence of plasma cells in the tested material. In addition, Tamer H. Said proved that 2D and 3D ultrasonography performed on infertile women at two phases of the menstrual period can predict the presence of chronic endometritis as a subtle cause of infertility and might be an indication for hysteroscopic evaluation for these patients [22].

### 2.2. Inflammation Within the Uterine Endometrium

#### 2.2.1. The Endometrium

The uterine endometrium is a dynamic tissue characterized by cyclic changes orchestrated by hormonal fluctuations during the menstrual cycle [23]. Alongside its role in facilitating embryo implantation and supporting pregnancy, the endometrium is also subject to inflammatory processes integral to tissue homeostasis, immune surveillance, and defense against pathogens [24]. The dysregulation of endometrial inflammation has been implicated in various gynecological conditions, including endometriosis, adenomyosis, preeclampsia, and dysfunctional uterine bleeding, underscoring the importance of understanding the underlying mechanisms governing inflammatory responses within the endometrium [25,26,27,28,29].

#### 2.2.2. Inflammatory Response

Recent molecular genetic studies propose a revised model of endometriosis pathogenesis [30]. They suggest that epithelial progenitor or stem cells, responsible for regenerating uterine endometrium post-menstruation, can become hyperactive and trapped outside the uterus. These trapped cells then proliferate, forming nascent glands and recruiting surrounding stromal cells, leading to the development of deep infiltrating endometriosis. Once established, the ectopic tissue triggers chronic inflammation due to immune surveillance (this process will be further explained later). This inflammatory response, driven by nuclear factor-κB signaling, is aggravated by abnormalities in estrogen receptor-β and progesterone receptor pathways, influenced by local inflammation, thus creating a dysregulated inflammation-hormonal feedback loop [30].

A myriad of pro-inflammatory mediators, including cytokines, chemokines, prostaglandins, and matrix metalloproteinases, are involved in orchestrating inflammatory processes within the uterine endometrium [31]. These mediators are produced by various cell types within the endometrium, including stromal cells, epithelial cells, and immune cells, in response to physiological cues and environmental stimuli. Interleukin-1 (IL-1), tumor necrosis factor-alpha (TNF-α), and interleukin-6 (IL-6) are among the key cytokines implicated in endometrial inflammation, exerting pleiotropic effects on cellular proliferation, apoptosis, and immune cell recruitment [32]. Additionally, chemokines such as CXCL8 (IL-8) and CCL2 (MCP-1) play crucial roles in regulating leukocyte trafficking and immune cell infiltration within the endometrial microenvironment. Also, IL-33 levels are notably elevated in endometriotic lesions compared to healthy endometrium. In vitro, the stimulation of endometrial and endometriotic cells with IL-33 induced the production of pro-inflammatory and angiogenic cytokines. In a mouse model, IL-33 injections led to systemic inflammation, increased plasma pro-inflammatory cytokines, and enhanced vascularization and proliferation of endometriotic lesions. These findings highlight the role of IL-33 in promoting inflammation, angiogenesis, and lesion progression in endometriosis [33].

#### 2.2.3. Immune Response

The uterine endometrium harbors a diverse array of immune cells, including macrophages, natural killer (NK) cells, T lymphocytes, and dendritic cells, which collectively contribute to immune surveillance and tissue remodeling during the menstrual cycle and pregnancy [34]. Macrophages, in particular, exhibit phenotypic plasticity and can adopt pro-inflammatory (M1) or anti-inflammatory (M2) states in response to local cues, thereby modulating the balance between inflammation and tissue repair [35].

NK cells (natural killer cells) play pivotal roles in regulating vascular remodeling and trophoblast invasion during early pregnancy, while T lymphocytes contribute to immune tolerance and defense against pathogens within the endometrial milieu [36,37]. The dysregulation of immune cell function within the endometrium has been implicated in various pathological conditions, including recurrent pregnancy loss and endometrial cancer, highlighting the intricate interplay between immune responses and endometrial homeostasis [38,39]. In endometriosis, immune cells in the endometrium behave differently, with a predominance of pro-inflammatory macrophages and abnormal activity in natural killer cells. Conflicting data arise from small studies with varied hormonal contexts and methodologies. Phenotyping immune cell subtypes is crucial for understanding their role in reproductive health and developing targeted treatments [40].

Different leukocytes also exhibit specific changes within the human endometrium across the menstrual cycle. Leukocyte numbers increase notably during the secretory phase, comprising around 40% of stromal cells premenstrually, primarily uterine natural killer cells. Macrophages, mast cells, dendritic cells, neutrophils, eosinophils, and regulatory T cells also play key roles in menstruation [41]. Progesterone withdrawal premenstrually boosts inflammatory mediators like IL-8 and MCP-1, driving leukocyte recruitment. Macrophages and neutrophils contribute defensins and whey acid protein motif proteins for microbial protection during epithelial barrier disruption. Mast cells activate closer to menstruation, triggering matrix metalloproteinases and extracellular matrix degradation [41]. Dendritic cells and macrophages help clear cellular debris from the uterine cavity, reducing viable cellular material in the Fallopian tubes, regulated by regulatory T cells. These processes are pivotal in endometrial repair mechanisms. Recent evidence also links immune cell disturbances and cytokine mediators to abnormal uterine bleeding and pelvic pain, underscoring the inflammatory process’s significance in both normal and abnormal endometrial bleeding [41].

#### 2.2.4. Menstruation

Menstruation exhibits inflammatory characteristics, with the intricate process of tissue breakdown and bleeding gradually being elucidated. Progesterone’s decline in the late secretory phase triggers interdependent inflammatory events within the endometrium. Loss of progesterone prompts decreased prostaglandin metabolism and vulnerability to reactive oxygen species (ROS), leading to NFκB activation and increased synthesis of pro-inflammatory factors such as prostaglandins, cytokines, and matrix metalloproteinases (MMP). This cascade recruits leukocytes, enhancing tissue degradation and bleeding. Simultaneously, microenvironmental changes induce neutrophils and macrophages to shift towards anti-inflammatory phenotypes, aiding rapid tissue re-epithelialization and integrity restoration [42,43].

#### 2.2.5. Implantation

In the endometrial secretory phase, fluctuations occur in the expression of cytokines, growth factors, and adhesion molecules, facilitating implantation. These immune components are intricately regulated by oestradiol and progesterone. Emerging research indicates that implantation and early pregnancy entail a pro-inflammatory environment. Supporting this notion is the observation that local injury aids implantation [44].

Emerging evidence suggests that aberrant inflammatory signaling within the uterine endometrium may adversely affect reproductive outcomes, including implantation failure, miscarriage, and infertility [32]. Human implantation may be described as the complex interplay between endometrial receptivity, inflammation, and the microbiome [44].

#### 2.2.6. Endometritis

Chronic endometrial inflammation has been associated with impaired endometrial receptivity, leading to suboptimal embryo implantation and subsequent pregnancy loss. For instance, inflammation plays a vital role at the site of implantation. Cytokines, growth factors, and immune cells work together to facilitate decidualization and allow implantation to occur [44]. While previous research has emphasized the association between embryo implantation and an active Th1 inflammatory response, alongside the requirement for a Th2-humoral inflammation for pregnancy maintenance, recent discoveries propose an additional necessity for a Th1 inflammatory response in acquiring uterine receptivity [45].

The condition of anti-inflammatory response may have a major impact on reproductive performance in older mammals. Uterine aging may also modulate oxidative stress, anti-inflammatory response, inflammation, mitochondrial function, and DNA damage repair; however, it could be potentially modulated by pharmacological agents, such as quercetin and dasatinib [46]. Understanding the molecular mechanisms underlying endometrial inflammation holds promise for the development of novel therapeutic interventions aimed at restoring endometrial homeostasis and improving reproductive outcomes in women affected by these conditions [47]. Figure 2 depicts changes in the endometrial microenvironment during menstrual cycle.

### 2.3. In Vitro Fertilisation (IVF) Failure

#### 2.3.1. IVF Failure Due to Inflammation

A significant cause of infertility is inefficient implantation, of which chronic endometritis is the main cause. It is, therefore, important to be able to accurately diagnose even the slightest signs of chronic endometritis [48]. Current diagnostic procedures include hysteroscopy and histopathological evaluation of an endometrial biopsy specimen. The main population on which the diagnosis of chronic endometritis is based is the detection of plasma cells in endometrial tissue. Plasma cell detection is usually based on CD138 immunohistochemical staining for more specific detection in double staining combination with nuclear marker MUM1-IRF. Even the presence of single plasma cells at 10 high-power fields is significant. In some studies, the rate of chronic endometritis in women with infertility problems has even reached 30–43% in the IVF group of women, dropping after antibiotic and probiotic therapy to 10–11% [48,49].

#### 2.3.2. Plasma Cells Detection in Infertility and Recurrent Implantation Failures in IVF

Infertility is a common gynecological disease. In recent years, the number of patients suffering from any form of infertility has been increasing worldwide, and even 1 in 6 people globally may be affected by the infertility problem [3,50]. Repeated implantation failures and recurrent pregnancy loss can be caused by endometriosis and chronic endometrial inflammation. Repeated implantation failures are a challenge to overcome in IVF treatment, although there are no official diagnostic criteria. The reason for repeated implantation failures can be either maternal or embryonic, but no targetable details are currently known [51].

NK cells and plasma cells (CD38+, CD138+), a characteristic feature of chronic endometritis, are supposedly linked with infertility and endometriosis-associated infertility [1]. A cohort study from 2016 performed on 93 patients planning to undergo assisted conception treatments showed that detection of CD138 in immunochemistry in the endometrial tissue can improve the diagnosis of chronic endometritis and, as such, should be advised in case of miscarriages and infertility [3]. Figure 3 depicts endometrium with plasma cells (an inflammatory process) and without plasma cells (no inflammatory process).

Because the definition of chronic endometritis differs between studies, it is worth examining how many plasma cells are enough to present the clinical outcome of chronic endometritis and cause fertility issues [2,52]. A retrospective study published in 2021 was performed on 716 infertile patients who never underwent either chronic endometritis, analysis, or antibiotic treatment. The number and distribution of CD138+ cells were analyzed by immunohistochemistry in the endometrial scratch. The study confirmed that CD138+ cells are a reliable method for identifying chronic endometritis. The authors suggested that chronic endometritis can be detected when the presence of plasma cells is ≥5 in at least one out of 30 high-power fields (HPF). In the meta-analysis also, a threshold of ≥5 plasma cells in 10 HPFs has been suggested for the diagnosis of chronic endometritis, and below 5 plasma cells in 10 HPFs may not determine chronic endometritis nor predict pregnancy outcomes [2,52]. In another clinical trial, as many as 145 women with unexplained infertility were investigated with the aid of hysteroscopy and immunohistochemistry for the presence of CD38 and CD138 marker cells. 75 (51.7%) of them suffered from chronic endometritis (the difference between control and analyzed groups—patients with definite infertility causes—was statistically significant). In addition, the authors suggested that patients with undefined infertility should be timely tested (hysteroscopy and immunohistochemistry) for the presence of CD38 and CD138 markers as plasma cells in order to exclude chronic endometritis [53]. Figure 4 shows the endometrium of a patient with chronic endometritis, having a significant amount of plasma cells.

The presence of CD138+ cells (and the diagnosis of chronic endometritis based on that) was also noticed in patients struggling with repeated implantation failures. Chronic endometritis is considered to reduce endometrial receptivity and so—has a negative impact on IVF outcomes [54,55]. A retrospective study on 126 repeated implantation failure patients (who were neither diagnosed with chronic endometritis nor received treatment) showed that 11.9% of repeated implantation failure patients suffered from chronic endometritis—the diagnosis was established based on the presence of five or more CD138+ cells in HPF [54]. Chronic endometritis seems to be an important cause of repeated implantation failures, and it is recommended to perform a chronic endometritis-associated examination, for example, immunostaining. Especially the treatment of chronic endometritis is easily accessible and brings promising results. In the research published last year, there were 327 patients observed from 2018 to 2020. A total of 117 of them suffered from chronic endometritis, which was confirmed with immunohistochemical staining (MUM-1+/CD138+); 89 patients were strongly positive, and 28 patients were weakly positive. Patients with chronic endometritis underwent treatment with doxycycline hydrochloride (Jiangsu Lianhuan Pharmaceutical Co., Ltd. at No.9, Jiankangyi Road, Biological Health Industry Park, Yangzhou, Jiangsu, China, 100 mg, op, Bid, 14 days) and metronidazole (Gold Day Pharmaceutical Co., 500 mg, op, Tid, 14 days) and the staining was performed again. Seven patients remained strongly positive, three became weak positive, and sixty patients became negative [55].

## 3. Conclusions and Future Directions

On the basis of current increasing trends, in the near future, the percentage of infertile couples may further skyrocket, leaving a lot of room for improvements and new technologies harnessing. With the goal of increasing the diagnosis rate of chronic endometritis, routine CD138 immunohistochemical staining should be considered in women with increased risk factors for chronic endometritis. These include prolonged episodes of menstrual bleeding, a previous abortion or miscarriage history, and fallopian tube obstruction. The CD138 immunohistochemical staining may clarify the diagnosis and enable timely treatment that will improve the pregnancy rate in patients [3]. A study by Angel Santoro et al. confirmed that plasma cells (CD138+) are a reliable method for identifying chronic endometritis. The authors proposed a scheme for diagnosing chronic endometritis when the presence of plasma cells is ≥5 in at least one of 30 high-power fields (HPF). It has also been suggested that a threshold of ≥5 plasma cells in 10 HPFs for the diagnosis of chronic endometritis and less than 5 plasma cells in 10 HPFs may not define chronic endometritis or predict pregnancy outcome [2]. This method seems to be a quick and cost-effective way that may increase the IVF success rate significantly.

It has been proven that dysregulation of chronic inflammation of the uterine endometrium is the basis of gynecological disorders such as endometriosis, adenomyosis, preeclampsia, and dysfunctional uterine bleeding. Understanding the basic mechanisms regulating inflammatory reactions in the endometrium may provide a basis for explaining the occurrence of these disorders [25,26,27,28,29]. Microflora of the uterine endometrium may be one of such factors. In search of the etiological factor of chronic endometritis, 16S ribosomal RNA (rRNA) sequencing analysis was performed for the microflora of the uterine endometrium. The results showed a relative dominance index of *Lactobacillus iners* and a positive percentage of *Ureaplasma* species, which suggests that these pathogens may be responsible for the etiology of chronic endometritis [12].

Another factor involved in endometritis may be the presence of polyps. Extensive research should be conducted on the prevalence of endometrial polyps in premenopausal women previously diagnosed with chronic endometritis. Randomized, controlled study should also be conducted in women undergoing hysteroscopic polypectomy to evaluate the role of antibiotic therapy against chronic endometritis in order to improve the success of IVF cycles and prevent endometrial polyps’ recurrence in the long-term.

We should also highlight the regenerative potential of endometrial stem cells [23]. The regenerative capabilities of stem cells found in the endometrium of the uterus are comparable to the regenerative capabilities of stem cells found in the bone marrow. The uterine endometrial stem cell population is heterogeneous and complex. Very advanced single-cell analysis technologies are increasingly used to examine it, such as single-cell ATAC sequencing (scATAC-Seq), single-cell RNA sequencing (scRNA-Seq), and spatial transcriptomics. Discovering and understanding detailed molecular mechanisms regulating the formation of chronic endometritis gives hope for the assisted fertilization techniques of new targeted therapeutic methods aimed at restoring endometrial homeostasis and may constitute a breakthrough in the treatment of infertility.

Overall, we assume that patients dealing with infertility, miscarriages, or recurrent implantation failure after IVF should be advised to have a CD138 immunohistochemical staining examination, which remains a very rapid and cost-effective procedure. This will enable clarifying their diagnosis and preparing the best possible treatment option which will improve their reproductive prognosis [1,3]. Despite the fact that both the inflammation within the uterine endometrium and the plasma cells’ role in IVF failure are under investigation, this additional test performed before the IVF procedure might be highly informative and provide a clue to the pre-IVF antibiotic therapy, possibly resulting in higher chances of implantation success.

## Figures and Tables

**Figure 1 ijms-25-13083-f001:**
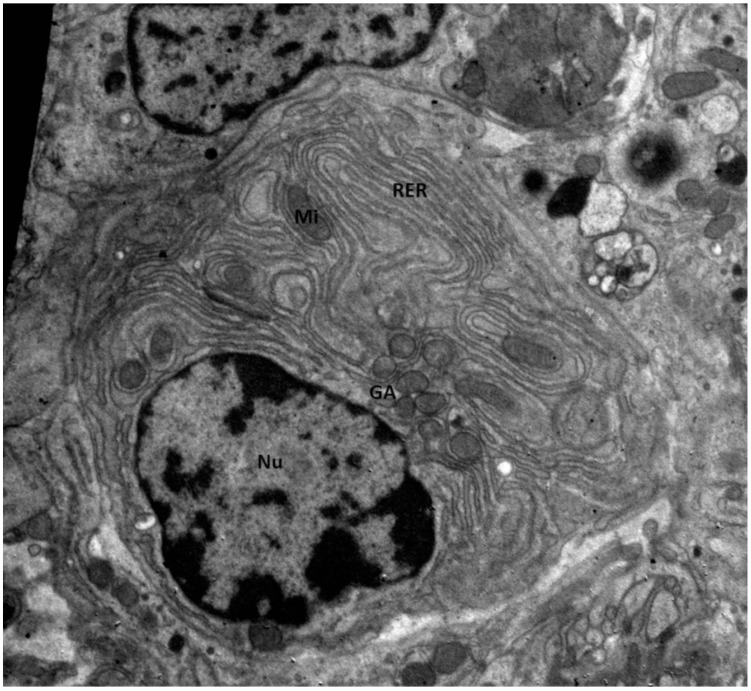
Plasma cell seen under the electron microscope; nucleus—Nu, rough endoplasmatic reticulum—RER, Golgi Apparatus—GA, mitochondrium—Mi. Material fixed in a fixative according to Karnowski’s protocol, embedded in epoxy resin, and prepared for electron microscopy evaluation on copper grids contrasted with a 9% saturated uranyl acetate solution and colored additionally with lead citrate; magnification: 5000× [source of photographs: authors’ own archive].

**Figure 2 ijms-25-13083-f002:**
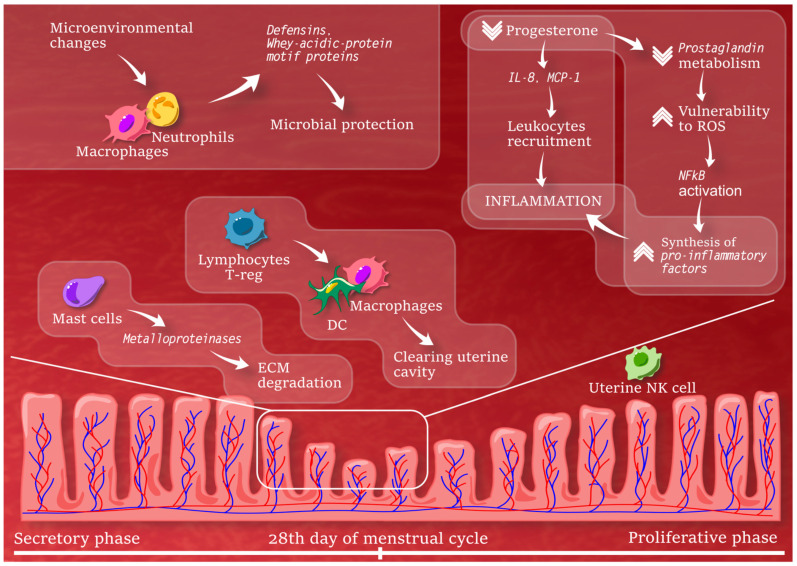
Endometrium with microenvironmental changes during secretory phase bearing mast cell engagement, and the role of NK cells and macrophages in the replacement and growth phase of endometrium; IL-8—Interleukin 8, MCP-1—Monocyte chemoattractant protein 1, ROS—Reactive oxygen species, NFkB—Nuclear factor kappa-light-chain-enhancer of activated B cells, ECM—Extracellular matrix, DC—Dendritic cell, NK—Natural killer.

**Figure 3 ijms-25-13083-f003:**
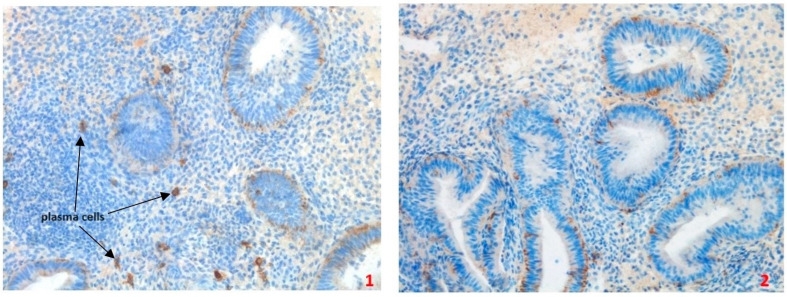
Endometrium with and without inflammatory, indicating plasma cells. Slide 1—CD138 (positive reaction) plasma cells marker, surface membrane staining. Tissue slide 1 presents an early secretory phase of the endometrial cycle with plasma cells within and around the endometrial lymphoid follicle. Slide 2—CD138 (negative reaction) plasma cells marker, normal early secretory phase of endometrial tissue with no signs of inflammation. Immunohistochemical reactions were performed according to the manufacturer’s protocol (CD138/syndecan-1 mouse monoclonal antibody, Cell Marque Corporation USA, 6600 Sierra College Blvd, Rocklin, CA 95677) using a Benchmark ULTRA device (Ventana Medical Systems, Inc., 1910 E. Innovation Park Drive, Tucson, Arizona 85755, USA). Magnification for all slides 200×. [source of photographs: authors’ archive].

**Figure 4 ijms-25-13083-f004:**
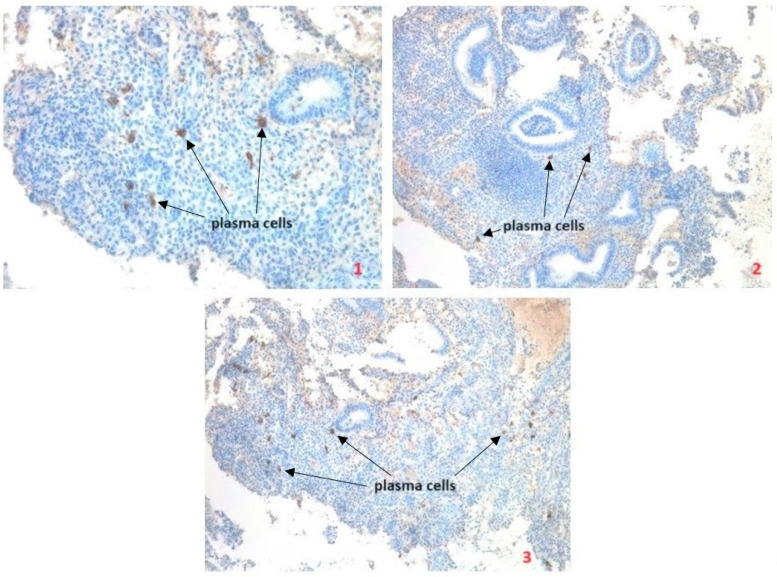
Endometrium of a patient with chronic endometritis, with a significant amount of plasma cells (1, 2, 3). IHC staining: CD138 (positive reaction)—plasma cells marker, surface membrane staining. Immunohistochemical reactions were performed according to the manufacturer’s protocol (CD138/syndecan-1 mouse monoclonal antibody, Cell Marque Corporation USA, 6600 Sierra College Blvd, Rocklin, CA 95677) using a BenchMark ULTRA device (Ventana Medical Systems, Inc., 1910 E. Innovation Park Drive, Tucson, Arizona 85755, USA). Magnification for slide 1 is 400×, for slides 2 and 3:200×. [source of photographs: authors’ archive].

## Data Availability

No new data were created or analyzed in this study. Data sharing is not applicable to this article.

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
