# Peer review of "Plasma Cells as the Key Players of IVF Failure? Unlocking the Enigma of Infertility and In Vitro Fertilization Failure in the Light of Uterine Inflammation"

_ijms, 2024, doi:10.3390/ijms252313083_

Round 1

Reviewer 1 Report

Comments and Suggestions for Authors

First, I would like to thank you for the invitation to review this overview paper.

The topic addressed by the authors is very relevant. The introduction is good and sufficiently informative. In the following sections, where the authors discuss endometrial inflammation and inflammation in IVF, there is no clear distinction made between the mechanisms of inflammation and the inflammatory mediators in cases of specific and nonspecific infectious inflammation of the uterus versus autoimmune inflammation, also from a pathohistological perspective. The conclusion is overly general, without clear guidelines for treatment.

Author Response

Dear Reviewer,

Thank You very much for Your time and effort, as well as constructive comments concerning our manuscript. We have studied Your comments carefully, had team discussions :) and made few significant corrections and improvements, which we hope meet with Your approval. Thanks to this work being done, we were also able to expand our knowledge even more - and enhance our future projects.

We decided to highlight all added or changed paragraphs (red text colour or highlighter) to facilitate the process of reviewing and save Your time.

We really hope theses modifications can meet with Your approval.

Thank You very much for Your time and effort. We know that nowadays time is scarce. 

 Yours Sincerely,

Authors

Reviewer 2 Report

Comments and Suggestions for Authors

This manuscript provides an overview of the role of plasma cells and inflammation in endometritis and infertility. The topic is certainly important, but there are also several issues in the manuscript that need clarification.

Lines 45-46: please check this sentence, the content is confusing.

There are repeated sentences in the manuscript, for example in lines 94-95 and 114-115.

How is chapter 2.1. related to the rest of the manuscript? It is as if something is being introduced, but the findings of the ultrasonography are not described further in the context of the plasma cells and inflammation.

Chapter 2.2 is very long; the authors could consider whether they could make some subheadings there.

Figure 2: do the highlighted processes occur exclusively around 28 days, just before menstruation? Also, if this figure was less red, it would be better to look at.

Lines 152-163: Turner et al. 2012 is cited to several times, the content of the paragraph does not quite match the referenced article. In any case, it is not possible to find the facts presented here.

Lines 185-194: this entire section has one reference at the end. But only the word MMP seems to link this text with this reference?

Line 232: this reviewer dares to doubt that CE is the primary cause of implantation failure.

Figures 3 and 4: arrowheads do not seem to point exactly to plasma cells in all cases.

Line 272 mentions a study published in 2020, but the reference is missing. This sentence is also a bit unclear.

The text on lines 303-313 would need rewording to be more readable.

Lines 276-280 and 325-329 repeat the same content.

Line 343: is endometriosis correct or should it be endometritis?

Lines 349-358: the meaning of this section is unclear. How are stem cells and endometritis related?

Lines 315-367: this paragraph should be more specific, so that it is clear what the summary of the given topic is and what the future directions are.

Author Response

Dear Reviewer,

Thank You very much for Your time and effort, as well as constructive comments concerning our manuscript. We have studied Your comments carefully, had team discussions :) and made few significant corrections and improvements, which we hope meet with Your approval. Thanks to this work being done, we were also able to expand our knowledge even more - and enhance our future projects.

Probably the most problematic element was the stem cells of the endometrium. There are several interesting papers about this topic. We know for example, that not all of the reparative potential of endometrial stem cells are related to their proliferation and differentiation features, but also their other properties such as immunomodulatory capability, and such features can make them proper candidates in the treatment of some autoimmune associated degenerative diseases like MS or CNS inflammation.  Endometrial stem cells were shown to have the potential to be ‘off the shelf’ clinical reagents for the treatment of heart failure. But this is of course the matter of future experiments. 

We decided to highlight all added or changed paragraphs (red text colour or highlighter) to facilitate the process of reviewing and save Your time.

We really hope theses modifications can meet with Your approval.

Thank You very much for Your time and effort. We know that nowadays time is scarce. 

 Yours Sincerely,

Authors

Round 2

Reviewer 1 Report

Comments and Suggestions for Authors

The authors have made changes in accordance with the suggestions. I believe the paper should be published.

Author Response

Dear Reviewer, 

thank you very much for your time and effort, 

have a good day, 

yours sincerely - The authors

Reviewer 2 Report

Comments and Suggestions for Authors

Some improvements have been made, but some issues still need to be clarified.

Section 2.1.: The authors have cited a 30-year-old paper as an example here (Narayan et al. 1993) and mentioned that it could be used as a basis for the development of confirmatory tests for endometritis. It does need some further clarification though, has it proven useful in 30 years? If so, it should be possible to find a more recent article about it.

The Turner et al. 2012 reference has now been removed from the text, but remains in the reference list.

Li et al. 2020 is now cited to in the lines 282 and 291, but such an article is not in the reference list.

Some paragraphs in the manuscript are marked in red, as if something had been changed, but the text seems to be word for word the same as in the previous version (e.g. lines 85-91, 354-369). Has there been a plan to change something, but it has not been done?

And yet I do not find in the text the changes which are said to have been made (e.g. Information and the references has been amended, lines 254 – 256).

Author Response

Dear Reviewer,   thank you very much for your attention and hints, we've tryied to amend every step. Especially:    Section 2.1.: The authors have cited a 30-year-old paper as an example here (Narayan et al. 1993) and mentioned that it could be used as a basis for the development of confirmatory tests for endometritis. It does need some further clarification though, has it proven useful in 30 years? If so, it should be possible to find a more recent article about it.

Indeed, if the knowledge has not changed and it is still valid, the paper provides a good background on the topic. But to make it more reliable we've added some latest papers on the topic to the manuscript - in this case from 2021. 

The Turner et al. 2012 reference has now been removed from the text, but remains in the reference list. Removed, thank you.    Li et al. 2020 is now cited to in the lines 282 and 291, but such an article is not in the reference list. Removed and corrected, thank you.

Some paragraphs in the manuscript are marked in red, as if something had been changed, but the text seems to be word for word the same as in the previous version (e.g. lines 85-91, 354-369). Has there been a plan to change something, but it has not been done? Several changes have been made during the process of revisions, from the very begining of the process, receiving important tips and hints from both Reviewers, thus, we've kept the changes visible from the moment we'd received an information we should track and colour all changes we do. 

And yet I do not find in the text the changes which are said to have been made (e.g. Information and the references has been amended, lines 254 – 256). During the revision process we've amnded several things in the mansucript and lines have been changed significantly many times, as a result of our work. All changes have been implemented, however we will remember to check the lines and and all the details while implementing corrections in the future, thank you very much for your important remark. We'll double-check the references before the paper is published, and we've also decided to change the program helping with references - this time we'd been trying two of them, because some people prefer to use Mac, others - Windows :) Thank you for understanding.